# Aging in mice alters regionally enriched striatal astrocytes

Kay E. Linker [1,2,4] ✉, Violeta Duran-Laforet[3], Matthias Ollivier[1], Xinzhu Yu [1,5], Dorothy P. Schafer[3] & Baljit S. Khakh [1,2] ✉

Aging affects multiple organs and within the brain drives distinct molecular changes across different cell types. The striatum encodes motor behaviors that decline with age, but our understanding of how cells within the striatum change remains incomplete. Using single-cell RNA sequencing from young and aged mice we identify molecularly distinct astrocyte subtypes. We show that astrocytes change significantly with age, exhibiting downregulation of genes, reduced diversity, and a shift to more homogenous inflammatory transcriptomic profiles. By exploring where striatal astrocyte subtypes are located with single-cell resolution, we map astrocytes enriched in dorsal, medial, and ventral striatum. Age increases inflammatory marker transcripts in dorsal striatal astrocytes, which display greater age-related changes than ventral striatal astrocytes. We impute molecular interactions between astrocytes and neurons and find that age particularly reduced interactions related to *Nrxn2*. Our data show that aging alters regionally enriched striatal astrocytes asymmetrically, with dorsal striatal astrocytes exhibiting greater age-related molecular changes.

The striatum is a large, evolutionarily conserved brain nucleus that serves multiple essential functions[1], including precise information encoding necessary for a range of motor and reward-associated behaviors[2,3]. Normal aging disrupts striatal function, resulting in degradation of motor learning skills and susceptibility to age-related neurodegenerative diseases such as Huntington's disease and Parkinson's disease[4–8]. Furthermore, owing to its limited regenerative capacity, the brain may be particularly susceptible to the accumulating damage that occurs with age[9], contributing to dysfunctional circuits and behavioral decline[10]. Non-neuronal glial cells are essential to support neurons and neural circuits[11], and exhibit age-related changes. Thus, RNA sequencing of several brain regions, including the striatum, demonstrates that age alters the molecular signatures of glia more robustly than those of neurons[12–14]. In mice, several transcriptomic approaches with single-cell resolution show that glia have more

pronounced age-related changes in gene expression and cellular density than neurons[14,15].

Astrocytes are glial cells that tile the central nervous system (CNS), displaying distinct morphologies and molecular signatures across the CNS[16–20]. Within the striatum, astrocytes contribute to normal and maladaptive behaviors by regulating neurons through a variety of mechanisms[21–31]. Using Rpl22-HA ribosomal subunits and the *Gfap* promoter to target astrocytes, gene expression analyses of astrocytes within several brain regions (visual and motor cortex, hypothalamus, and cerebellum) demonstrated age-induced shifts in their transcriptomes[32]. Notably, a consistent elevation in inflammatory transcripts with unique patterns of decline was observed in individual brain regions[32]. In a separate study, RNA sequencing of striatal astrocytes from the *Aldh1l1*-eGFP-L10a mouse line expressing GFP-tagged L10a ribosomal subunits also demonstrated increased inflammatory

[1]Department of Physiology, David Geffen School of Medicine, University of California, Los Angeles, Los Angeles, CA, USA. [2]Department of Neurobiology, David Geffen School of Medicine, University of California, Los Angeles, Los Angeles, CA, USA. [3]Department of Neurobiology, Brudnick Neuropsychiatric Research Institute, University of Massachusetts Chan Medical School, Worcester, MA, USA. [4]Present address: RefinedScience, Aurora, CO, USA. [5]Present address: Center for Neuroimmunology and Glial Biology, The Brown Foundation Institute of Molecular Medicine for the Prevention of Human Diseases, University of Texas Health Science Center at Houston, Houston, TX, USA. ✉e-mail: kaylinker@gmail.com; bkhakh@mednet.ucla.edu

transcripts[33]. However, both these studies assessed astrocytes within individual brain regions as a pooled population because of the genetic approaches employed. Astrocytes are diverse between and within brain regions[16,17,19,34], and changes in any one brain region may not be reflective of changes elsewhere. It is also important to note that past studies did not evaluate age-related differences for astrocytes in different parts of the striatum, nor did they assess different populations of astrocytes in any brain region. Therefore, because of the methods available and employed in the aforesaid pioneering studies[32,33] an understanding of how striatal astrocyte subtypes[19,23,26] change with age remains unknown. This is an important issue to resolve because the striatum is involved in age-related neurodegenerative diseases that also exhibit dysfunctional astrocytes[35,36]. In addition, motor skill decline is a clinical metric of age, increasing mortality risk by 30%[37]. The striatum is a key regulator of such behaviors with well-established differences between dorsal-ventral areas, but the contributions of astrocyte subtypes to different areas of the striatum during aging remain unexplored. Considering these issues, we focused on assessing diverse striatal astrocytes and how they change with age by employing single-cell methods.

Single-cell RNA sequencing (scRNAseq) and spatial transcriptomics enable the discovery of previously unidentifiable cell types[38] and the ability to produce maps of all cell types within tissues. Much is known about the local heterogeneity and precise demarcation of neurons, but far less is known about striatal astrocytes, their local diversity, anatomical location, and their age-related changes. In this study, we used a series of integrated approaches to explore astrocytes within the striatum and determined how distinct astrocyte subtypes change with age in terms of their molecular properties within different anatomical locations.

## Results

### Astrocytes display distinct molecular changes with age

We used multiple integrated techniques to investigate the cellular and molecular changes that occur within the striatum between young (2 months) and aged mice (18 months; Fig. 1a), with a focus on astrocytes. For the scRNAseq data, we report quality control (QC) metrics before and after filtering steps (Supplementary Fig. 1), power analysis to assess if our data have enough cells to identify 7 astrocyte subtypes in the striatum (Supplementary Fig. 2), and cluster module scores for each astrocyte subtype for cell cluster integration and molecular distinction with the scRNAseq and multiplexed error-robust fluorescent in situ hybridization (MERFISH) data (Supplementary Fig. 3). Additional checks of quality and validation in relation to past work are included in the sections that follow and data are provided in Supplementary Data 7, Fig. 3g, and Fig. 5h-i

Generally, glia, and particularly astrocytes, are represented at low levels in standard scRNAseq protocols; we thus used published protocols that bias scRNAseq preparations for cellular diversity, enriching for glia at the expense of neurons[19,23,26] (Supplementary Figs. 4b). We assessed 64,836 cells in total from eight mice; the data for four 18-month-old mice were assessed along with data from four 2-month-old mice that we recently reported[23] (Methods). We recognize that greater numbers of cells would allow further insights to emerge, and we thus limit our conclusions to those that are supported by our dataset.

Astrocytes are of interest in the context of aging[39] and for exploring how astrocytes regulate striatal circuits[40] and neurodegenerative diseases[36], prompting us to analyze them in detail. Shallow clustering was used to identify large cell classes (Fig. 1b, c), from which major non-neuronal cells (astrocytes, oligodendrocytes, microglia, and endothelial cells) were subgrouped for a second tier of Louvain community detection (Fig. 1d, e; Supplementary Figs. 4 and 5). The astrocyte cell class exhibited known genes (Fig. 1c; e.g., *Slc6a11*). In accord with past work[19], we identified seven putative astrocyte

subtypes that could be split by molecular identity and age (Fig. 1d), and by cluster-defining genes (Fig. 1f). We also clarified that the cluster-defining genes are not categorical markers but instead embody sets of genes representing signatures used to define the cell clusters. We next compared cluster-defining genes (Fig. 1f) with published astrocyte gene expression data from deep sequencing of striatal astrocytes[20] and found 91% of these genes were detected. Thus, Fig. 1f shows the top cluster-defining genes (for A1–A7; Supplementary Data 3) and Fig. 1g shows their expression (as FPKM) from astrocyte RiboTag RNAseq data[20]. Furthermore, we compared the cluster-defining genes to recent scRNAseq data for striatal astrocytes[41] and found that all of them were detected (Fig. 1g). These evaluations show that cluster-defining genes for astrocyte subtypes A1–A7 seen in our data are in accord with astrocytic expression in two published datasets. Upon quantification (Fig. 1e), A1 and A5 subtypes emerged as dominant in aged mice, whereas A3, A4, A6, and A7 subtypes were dominant in young mice. Subtype A2 was equal between young and aged mice. We also found that astrocytes from aged mice displayed less molecular diversity than those from young mice; this was observed in striata that were clustered separately (Supplementary Fig. 4d [young = 6 clusters]; Supplementary Fig. 5c [aged = 3 clusters]) and in integrated aged/young datasets (Fig. 1d). Aged-astrocyte dominant subtype-defining transcripts (Fig. 1f) include *Stmn1*, a transcript whose overexpression distorts cell shape[42]; *Hes5*, a transcriptional repressor[43], and *Vgf* that is involved in inflammation[44]. One common subtype transcript (A2) is *Ddit4l*, a stress response gene that promotes autophagy[45]. Young astrocyte dominant subtype transcripts were more varied in function, including chaperoning small molecules (A3; *Trf, Hbb-bt*), transcriptional and proteome regulation (A4; *Rorb, Hspa2*), cell structure (A7; *Map1b, A6; Tnni1*), and calcium signaling (A7; *S100a4*)[46–49]. Overall, although all cell types displayed age-related changes in gene expression from our scRNAseq data, we found that the top 20 DEGs across cell types mapped to astrocytes and endothelial cells (Supplementary Fig. 6), which we evaluated in detail in the following sections with a focus on astrocytes.

### Comparison of scRNAseq to past work

There are two prior studies reporting how striatal astrocytes change with age in mice, which used *Aldh1l1* L10a RiboTag methods[33] and single-cell methods[14]. We next compared our findings at the single-cell level to the RiboTag study, both for validation purposes and to identify shared molecular features. In making these comparisons, it is important to note that RiboTag methods and scRNAseq differ in sequencing read depth. Nonetheless, we identified 926 common genes (Supplementary Fig. 7a, b). Of these, the top 16 significantly altered genes from our study (Log2FC > 0.25 and FDR < 0.05 for young versus old) and Clarke et al. (ref. 33 FPKM > 5 and FDR < 0.05 for young versus old) were shared (Supplementary Fig. 7b). We used Enrichr to perform pathway analysis, which highlighted commonalities in inflammatory reactivity (e.g., lysosomes, glutathione metabolism) and protein glycation (e.g., glycan degradation, glycosaminoglycan degradation). The prior study also concluded that aged striatal astrocytes adopt A1-like reactivity[33]. In accord, from our data, we found that striatal astrocytes from aged mice displayed broad changes shared with some of the previously reported pan reactive, A1 and A2 genes with 7, 3 and 3 DEGs, respectively (Supplementary Fig. 7c). While we did not find strong evidence for A1-like reactivity alone, upon closer examination we found that the data from Clarke et al.[33] also included changes in pan reactive, A1 and A2 genes with 7, 7 and 4 DEGs, respectively (Supplementary Fig. 7c in this study and Fig. 2E of ref. 33). These data suggest that aged striatal astrocytes adopt broadly reactive/inflammatory profiles that we found vary between striatal astrocyte subtypes (Supplementary Fig. 7d). The finding that astrocytes display neuroinflammatory and reactivity states with aging is a consistent finding between the current work and past RiboTag[33] and scRNAseq studies[14]. We next compared age-related astrocyte DEGs from our data to 170

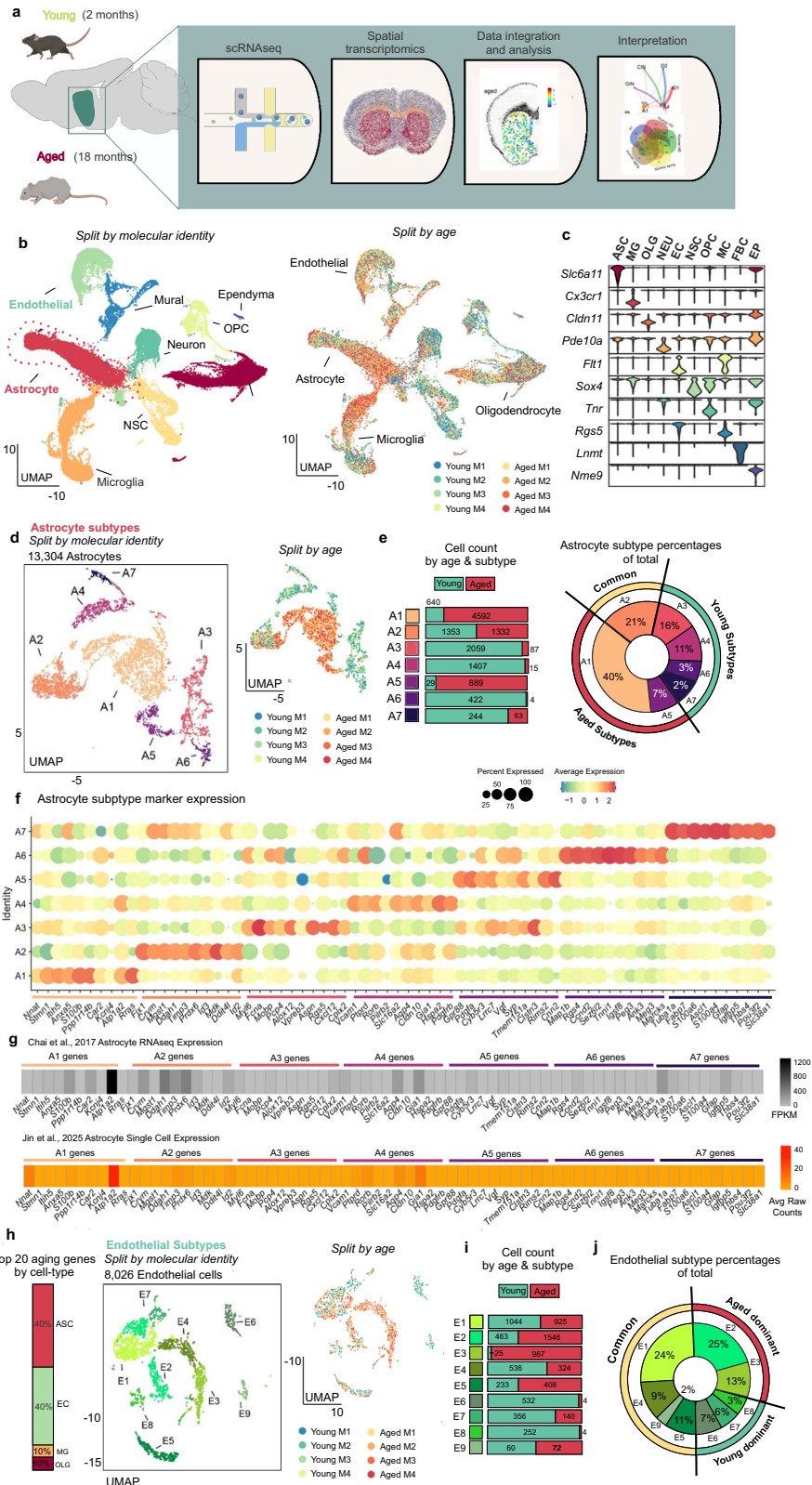

consensus astrocyte reactivity genes (cARGs) from multiple disease states[50]. Of the 170 cARGs, 88 genes were detected in our data, and of the detected genes 97% (85 out of 88) were significantly differentially expressed (42 up, 43 down), displaying age-dependent variation between astrocyte subtypes (Supplementary Fig. 7e, f). These analyses demonstrate overall correspondence between our data for aging striatal astrocytes and past work[33,50] and show that aged astrocytes

adopt inflammatory and reactivity states that vary between subtypes. Core conserved increases in select genes such as *C3*, *Gfap*, and *S100a6* were observed. However, we caution that methodological differences between our work and past studies[33,50] ought to be considered when making mechanistic interpretations of the similarities and differences between them. Furthermore, it is likely that additional age-related changes will be found in future studies that benefit from greater

**Fig. 1 | scRNAseq revealed differences between astrocytes from young and aged mice. a** Schematic of the approach to evaluate astrocytes within the striatum in young and aged mice (*n* = 4 mice per group, 8 total; 64,836 cells). The cartoons on the left side of (**a**) were created in Biorender (Linker, K. (2025) https://BioRender.com/4yka9a8). **b** UMAP split by age and molecular identity. **c** A violin plot of select markers for each major cell type (ASC astrocyte, MG microglia, OLG oligodendrocyte, NEU neuron, NSC neural stem cell, EC endothelial cell, OPC oligodendrocyte precursor cell, MC mural cell, EP ependyma cell) is shown (cross-referenced with dropviz.org) (**c**). The astrocyte cell class was subset from the larger dataset and analyzed further (**d–f**). **d** UMAP of astrocytes, which shows putative subtypes identified with Louvain clustering (resolution = 0.1), split by molecular identity and age. **e** The number of each cell type derived from young (green) or old (red) for each astrocyte subtype (A1–7) is shown to display the contribution of young or aged cells to each astrocyte subcluster. The relative percentage of each astrocyte subcluster commits to the total amount of astrocytes (**e**, inner pie) and if

the subcluster is dominated by cells from aged, young, or neither (common) samples (**e**, outer pie). Astrocyte subclusters were determined to be young or aged dominant if >70% of the cells within that cluster came from young or aged mice. Expression of 10 distinct subtype-defining markers for each subcluster is shown (**f**). **g** FPKM from the Chai et al.[20] RNAseq dataset is shown in the grayscale heatmap. The average raw count values of cluster-defining genes in the Jin et al.[41] single-cell sequencing of astrocytes are shown in the orange heatmap. **h** Astrocytes and endothelial cells contribute the highest amount of DEGs to the total aging DEGs across all cell types. **i** UMAP of subset endothelial cells have nine unique molecular subtypes that were identified with Louvain clustering (resolution = 0.1). **j** The number of each cell derived from young or old for each endothelial subtype (E1–9) is shown to display the contribution of young or aged cells to each endothelial subcluster. The relative percentage of each endothelial subcluster (**j**, inner pie) and if the subcluster is dominated by cells from aged, young, or neither (common) samples (**j**, outer pie).

---

numbers of high-quality cells, and our findings do not vitiate the necessity for such additional work.

## General comments in relation to other cell types

We note that our scRNAseq data provide information on non-neuronal cell transcriptomes more broadly beyond that of astrocytes alone (Supplementary Fig. 6). Other cell classes, including microglia, endothelial cells, and oligodendrocytes, exhibited several subclusters. Like astrocytes, endothelial cells displayed reduced molecular diversity with age (Fig. 1h; Supplementary Figs. 4g, 5f and 8c). In contrast, oligodendrocytes increased molecular diversity (Supplementary Figs. 4f, 5e and 8d). All three of these cell types displayed subclusters that are dominant in aged mice and those that are dominant in young. Common functional signatures threaded throughout the aged subtype markers include mitochondrial function (E5: *Alas2*; O4: *Slc16a1*), proteome regulation (E3: *Ctsd*; M6:*Kcdt12*; M9:*Ube2c*, O5: *Desi1*) and inflammation (M3: *Ptgs1, Pld4;* M4: *Cspg5;* M7:*Prf1*; E2: *Mgp;* O5: *Il33;* O7: *Igsf8*) (Supplementary Fig. 8). Based on these evaluations we conclude that striatal non-neuronal cells exhibit heterogeneity and that age manifests itself as shifts in their transcriptomes that are specific to each non-neuronal cell class. However, common functional signatures associated with age also emerge and are seen across non-neuronal cells (e.g., inflammation). These findings are consistent with a recent study[14].

## The regional distribution of striatal astrocyte subtypes changes with age

Our scRNAseq experiments demonstrate striatal astrocytes comprise separable subtypes that change with age (Fig. 1). We next chose one gene from each of the seven subtypes and performed RNAscope evaluations in striatal tissue sections to assess mRNA expression in S100β-positive astrocytes in 2-month-old mice. We used S100β as an astrocyte marker because it labels essentially all striatal astrocytes[23,24]. mRNA for all seven genes for clusters A1–A7, i.e., *Atp1a2, Crym, Cplx2, Aqp4, Pdgfa, Rgs4*, and *Gfap* were found in S100β-positive astrocytes within striatal tissue sections (Fig. 2a–g; 30–35 cells from 3 mice). As expected[20], *Gfap* was expressed at low levels in striatal astrocytes (Fig. 2g), but the RNAscope assessments also revealed that some transcripts varied significantly within astrocytes located in the dorsal, central, and ventral striatum (e.g., *Crym, Atp1a2* in Fig. 2). Next, to determine if astrocyte subtypes have separable spatial enrichment more broadly and to assess young and aged mice, we performed MERFISH[51]. MERFISH detects single RNA transcripts and provides single-cell resolution, which in our data encompassed 42,549 cells. We created a panel of 383 genes designed from our scRNAseq study (Supplementary Data 1). The genes we chose for the panel are categorized by three key features: known cell class markers, aging markers, and putative cell subtype markers. Known cell class markers included genes for major cell classes (astrocytes: *Aqp4*, neurons: *DRD1/DRD2*,

microglia: *Hexb*, see Supplementary Fig. 9). These markers were used to identify established cell classes for subsequent analysis. Aging-related markers were the top 20 age-induced DEGs across all cells. These markers were derived by performing MAST (Model-based Analysis of Single-cell Transcriptomics) differential expression analysis that treats cellular detection as a covariate. We were most interested in astrocytes and endothelial cells and targeted our panel for putative subtype transcripts that were cluster-defining genes for these cells from Fig. 1. We selected these genes by determining the ratio of the percent of cells inside the respective cluster to percent expression equal to or greater than 2 (e.g., if 50 percent of the cells inside the cluster express the marker, and 75 percent of the cells outside of the cluster do not; 0.5 expression within cluster divided by 0.25 expression outside of cluster = 2). We chose to select cluster-defining genes with a higher percentage of expression in each cluster, because a low percentage of cells within a cluster can have high expression of a gene, and drive FDR significance and increase logFC, but not be representative of the broader gene expression profile across the cluster. This ensured that a high proportion of the cells expressed the gene and that the cluster-defining gene was not driven by a small proportion of cells, providing a higher likelihood that we would capture this expression in our MERFISH experiment. These markers were all statistically significant after *p* value adjustment. For each subtype A1–7 and E1–9, the top 10 markers had an expression ratio greater than 2 and had a *p* value FDR < 0.05. The full list is provided in Supplementary Data 1. The cell class markers were used to identify major glia populations to create a cellular map of coronal sections that included the striatum (Fig. 3a). Cells located within the striatum were regionally segregated and clustered with Louvain community detection (Fig. 3a). Astrocytes of the striatum underwent a second tier of clustering mirroring scRNAseq analysis; several clusters were identified. The MERFISH data were also integrated with scRNAseq data using reciprocal principal component analysis (PCA) for collective comparison[52]. Both approaches identified seven astrocyte subtypes that are analogous to scRNAseq-identified subtypes (Fig. 3b). Each astrocyte subtype was identified by reliable markers that distinguished them both in MERFISH and in scRNAseq datasets (Supplementary Fig. 10).

Cell type-specific identifying markers mapped scRNAseq defined astrocyte subtypes to MERFISH astrocyte clusters, which enabled us to map astrocyte locations and regional enrichment (Fig. 3c-d; Supplementary Data 2). We assessed the striatum in 500 μm increments starting from the corpus callosum, down to the base of the striatum (NAc), for a total of six sub-regions along the dorsal-ventral axis. This astrocyte expression probability quantification was calculated by the number of astrocytes within a subcluster (A1–7), within each 500 μm subregion (0–5), divided by the total number of astrocytes within that section, and normalized to the total number of astrocytes within each respective subcluster. This quantification allowed us to calculate the percentage of astrocytes from a specific cluster within a subregion of

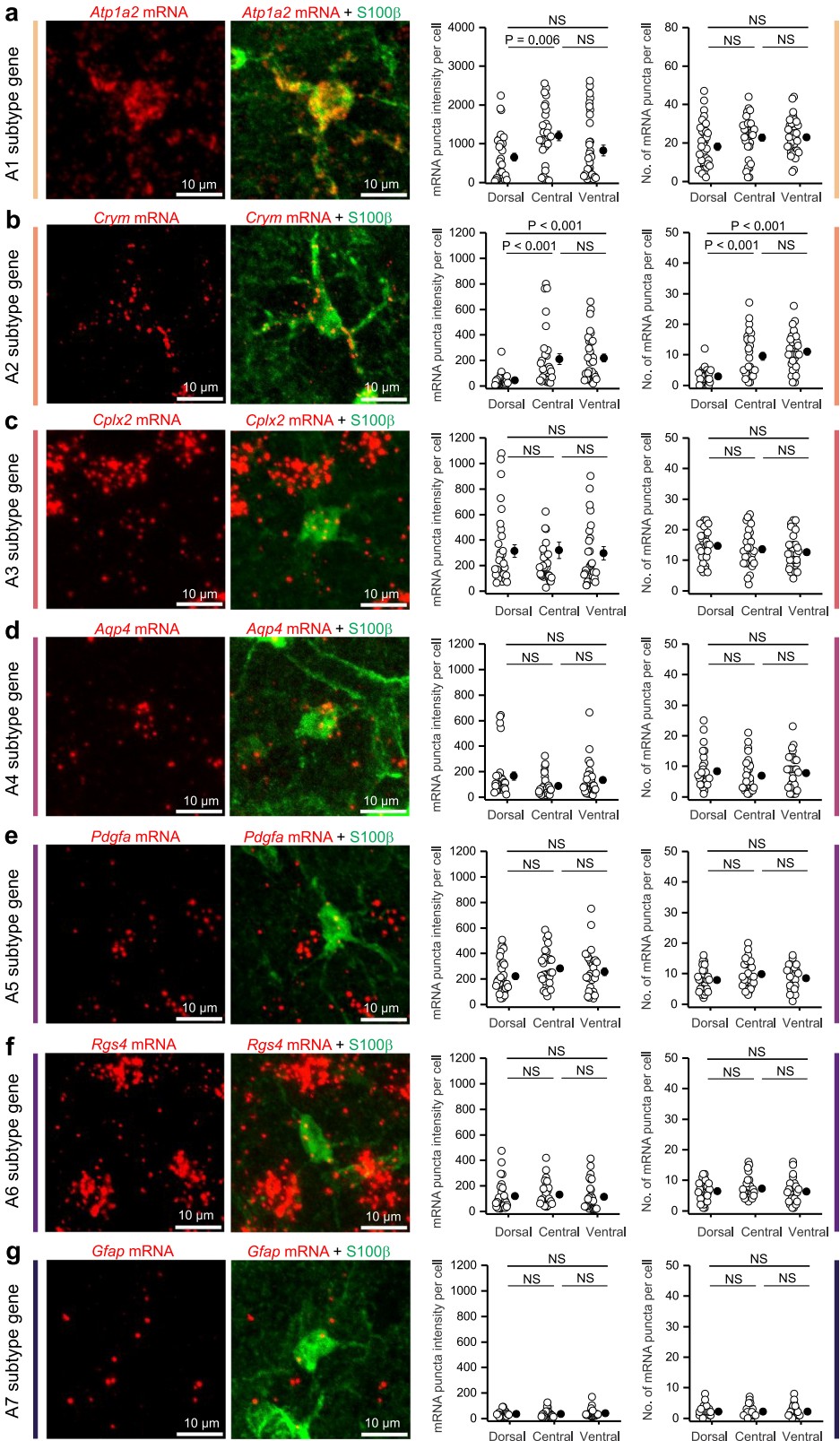

**Fig. 2 | RNAscope validation of candidate mRNA gene expression for each astrocyte subtype.** Representative images (left) of mouse central striatum with mRNA for *Atp1a2* (**a**), *Crym* (**b**), *Cplx2* (**c**), *Aqp4* (**d**), *Pdgfa* (**e**), *Rgs4* (**f**), and *Gfap* (**g**) in red, and S100β astrocyte marker protein in green. Quantification of the mRNA puncta intensity and the number of mRNA puncta per cell for each gene of interest in the dorsal, central, and ventral striatum (right). Average data shown as mean ± SEM. from *n* = 30–35 cells from 3 mice (one-tailed one-way ANOVA followed by Tukey's post hoc test or Kruskal-Wallis ANOVA). Source data are provided. In the case of *Pdgfa*, *Rgs4*, and *Cplx2*, mRNA was also detected in cells that were not S100β-positive astrocytes. These were NeuN-positive neurons. This is consistent with the idea that individual genes are not categorical markers but are part of a set of genes used to define cell clusters.

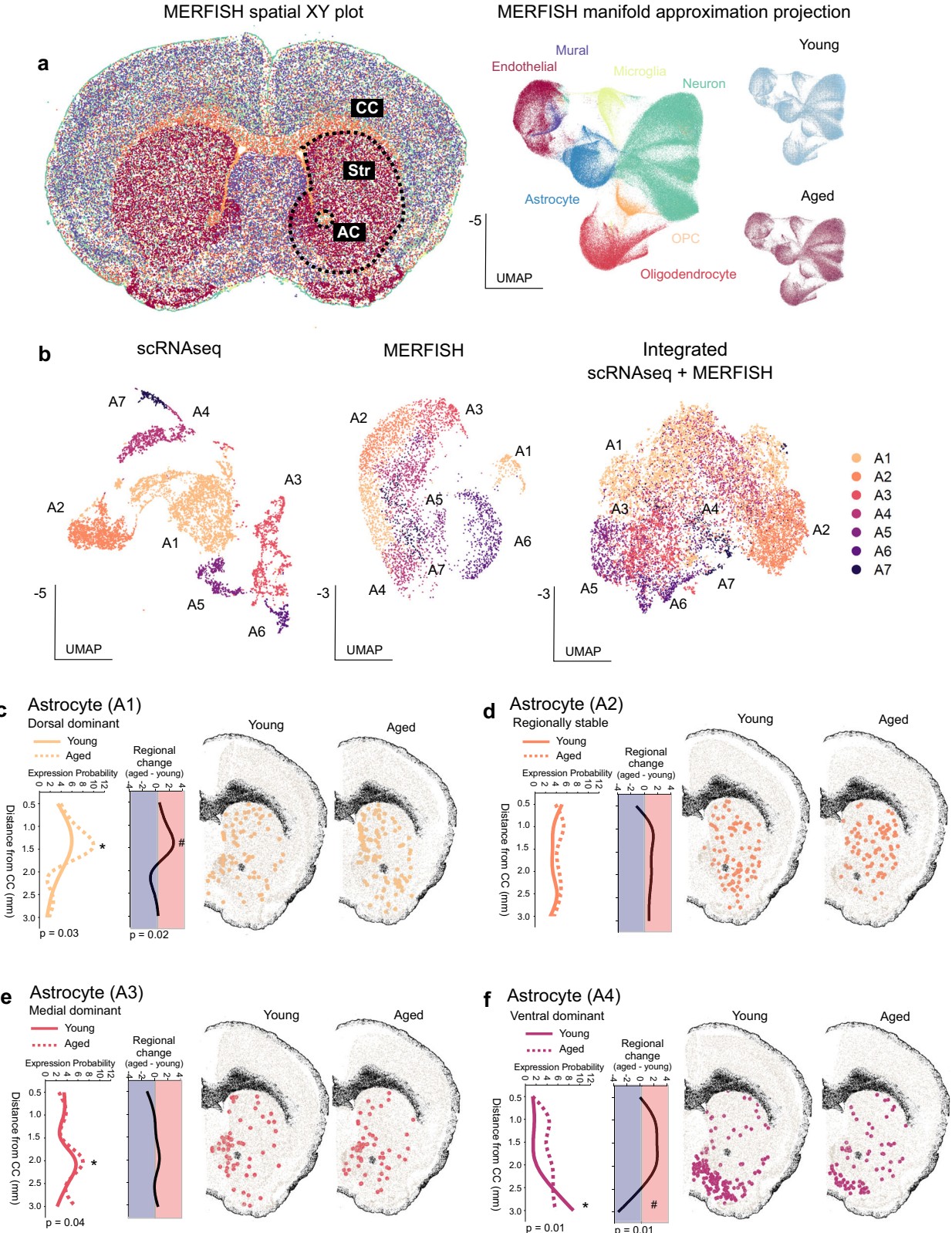

the striatum, without the confounding factors of different astrocyte subsets having more cells than others (A1 subsets have the most astrocytes of all the subsets) and different regions of the striatum display more astrocyte density than others (the ventral striatum has more astrocytes than the dorsal). This enabled us to quantify regional enrichment of astrocyte subtypes and their shifts or stability with age. Astrocyte subtype 1 expression was dominant in the dorsal striatum,

and in aged mice this dominance was exacerbated (Fig. 3c). Astrocyte subtype 2 was regionally stable in both young and aged mice (Fig. 3d). Astrocyte subtype 3 was dominant in the medial striatum in both young and aged mice (Fig. 3e). Astrocyte subtype 4 was dominant in the ventral striatum, and in aged mice this regional enrichment was diminished, with the enrichment shifting dorsally towards the corpus callosum (Fig. 3f). Some smaller astrocyte subtypes (5-7) also had

**Fig. 3 | MERISH shows that astrocytes from aged mice display different regional enrichment. a** XY directional map of all cell types identified within the striatal section is shown. Medium spiny neurons, cortical neurons, astrocytes, oligodendrocytes, endothelial cells, and microglia are displayed in different colors - each dot represents a cell segmented by MERLIN. All cells within the XY coordinates of the striatum were subset, and here we display the UMAP of these subset striatal cell clusters split by major cell types and UMAP split by age (young = blue; aged = red). **b** Striatal astrocytes were subset from all striatal cells and clustered separately using the Louvian algorithm. The first UMAP shows striatal astrocyte subtypes from single cells, the second UMAP shows striatal astrocyte subtypes from MERFISH, and the third UMAP shows striatal astrocyte subtypes from integrated single-cell and MERFISH datasets. **c–f** Top 4 astrocyte subtypes (by abundance) are shown. The astrocyte subtype expression probability is quantified along the dorsal-ventral axis in 500 μm segments in young and aged mice. We divided the striatum into five 500 μm sections, starting at the base of the corpus callosum and moving ventrally.

We quantified the density of each astrocyte subtype within each subregion. This astrocyte expression probability quantification was calculated by the number of astrocytes within a subcluster (A1–7), within each 500 μm subregion (0–5) ($X_{A1...A7}$ within $Y_{0...5}$) divided by the total number of astrocytes within that 500 μm section ($\Sigma_{total}$) normalized to the total number of astrocytes within each respective subcluster ($\sigma_{A1...A7}$) ([($X_{A1...A7}$ within $Y_{0...5}/\Sigma_{total}$)/$\sigma_{A1...A7}$]). The regional change is quantified by subtracting the young astrocyte expression probability from the aged expression probability. These quantifications were statistically analyzed using a two-way repeated measures (for subregion) ANOVA. Asterisks (*) indicate significant differences ($p$ value < 0.05) across sub-regions, and hashtags (#) indicate significant differences across ages. Individual representative astrocyte maps for young and aged striatal sections are displayed to the right of the astrocyte density quantification, with astrocyte subtypes demarcated in their respective colors. In the graphs shown in (**c–f**), the corpus callosum is abbreviated as CC on the y-axis.

regional enrichment; with A5 being dorsal dominant, A6 medial dominant, and A7 dorsal dominant (Supplementary Fig. 10). Thus, two transcriptomic approaches with single-cell resolution (scRNAseq, MERFISH) and their integration demonstrated heterogeneity of astrocytes within the striatum and showed that regional enrichment of astrocyte subtypes shifts with age, which is similar to a recent scRNAseq and MERFISH study[14]. Our data indicate that subtypes exhibit regional enrichment that varies with age. These changes are likely due to cells shifting their state during aging, rather than the emergence of new cells or the migration of existing ones. Our data provide a basis to understand these state shifts, but additional studies with far greater numbers of cells across different species are an important requirement for the future.

Endothelial subtype markers were included in our MERFISH panel. Clustering of MERFISH identified eight of the nine striatal endothelial cell subtypes both separately and when integrated with scRNAseq (Supplementary Fig. 11). We found that striatal endothelial cells also exhibited shifts in transcriptomic signatures in aged mice (Fig. 1h–j, Supplementary Fig. 8). However, in contrast to astrocytes, spatial mapping of endothelial cells did not demonstrate regional enrichment within the striatum (Supplementary Fig. 11). Thus, all striatal endothelial subtypes had dispersed densities within tissue, despite their molecular diversity at the scRNAseq level.

## Striatal astrocyte aging-related signatures

We investigated differentially expressed genes within astrocytes as a group in young and aged mice, outside of the astrocyte subtype analyses (Fig. 4a). Astrocytes and endothelial cells changed significantly with age, and most transcripts in astrocytes decreased with age, unlike other major cell classes within our single-cell dataset (Supplementary Fig. 6). Key transcripts that changed with age showed enriched expression in subtypes that had dominant expression in the dorsal striatum (Fig. 4a). For example, *Gfap* RNA increased with age, and astrocytes subtypes that occupied the dorsal striatum expressed more *Gfap* (Fig. 4b, Supplementary Fig. 12). GFAP is expressed at low levels in the striatum[19,20], but is known to increase with inflammation and aging, and is strongly associated with astrogliosis[53]. We measured the percent area of the striatal parenchyma covered by GFAP (in 500 μm²) and found low GFAP throughout the striatum in young mice (Fig. 4c, Supplementary Fig. 12). However, in aged mice, GFAP coverage was significantly higher in the dorsal striatum compared to young mice ($n = 4–5$ mice, $p = 0.015$) and dissipated along the ventral axis (Fig. 4c; $n = 4–5$ mice, $p > 0.1$). In addition, scRNAseq data showed *C4b* increased with age and marked aged-dominant astrocyte subtypes A1 and A5, and *Tubb3* decreased with age and marked young-dominant astrocyte subtypes A3 and A6 (Supplementary Fig. 13). MERFISH datasets demonstrated *C4b* transcripts increased primarily in the dorsal striatum with age, while *Tubb3* decreased more substantially in the ventral striatum (Supplementary Fig. 13).

Astrocyte number did not change with age in MERFISH or immunohistochemistry (IHC) datasets: S100β-positive astrocytes increased along the ventral axis from ~190 to ~240 cells per 500 μm² (Fig. 4d, e), but this pattern did not change with age. These findings suggest that age-related changes to astrocytes are not a result of astrocyte count decreases but due to transcriptomic shifts in state.

Major astrocyte aging pathways included activated neuroinflammation signaling, senescence, and deactivated synaptogenesis and stem cell pluripotency (Fig. 4f), which aligns with major pathways that are disrupted in other organs[54]. *Psen1* is an upstream regulator whose proteolytic cleavage is involved in Alzheimer's disease, a neurodegenerative disease with high comorbidity with age (Supplementary Fig. 13). *Rest*, which was activated, is also an upstream regulator and a major transcriptional repressor. This is consistent with the downregulation of astrocyte transcripts with age.

We created a composite aging score to investigate on a cell-by-cell basis which astrocytes within the striatum had more robust transcriptomic changes with age. The score is calculated by the average expression levels of the top 20 differentially expressed genes in age on a single-cell level, subtracted by the aggregated expression of control feature sets (the module score function in Seurat). Seurat selects a set of control genes to serve as a reference. The control genes are chosen randomly from the dataset but matched for expression levels to the genes in the input module. A positive module score in a cell shows that the module's genes are more highly expressed than the control genes in that cell. This visually delineates astrocytes by the top 20 aging DEGs across the dorsal-ventral axis of the striatum, with dorsal astrocytes having a higher aging score than ventral (Fig. 4g). This is consistent with recent findings showing specific age-related astrocyte states emerge near the corpus callosum adjacent to the dorsal striatum[14].

## Human and murine striatal astrocyte signatures in different contexts

To explore how the mouse striatum aging astrocyte transcriptome relates to past work, we compared A1 astrocyte striatal astrocyte subset enriched genes to four groups: (1) mouse aging astrocyte DEGs (across all astrocyte subtypes), (2) human aging transcripts enriched in astrocytes[12] (microarray), (3) Huntington's disease (snRNAseq)[55], and (4) Parkinson's disease (snRNAseq)[56] (Fig. 4h, i). The total number of overlapping DEGs and top 10 transcripts (by adjusted $p$ value) between each of these conditions, with the A1 subtype cluster-defining genes, are shown as Venn diagrams and heatmaps, respectively, in Fig. 4h, along with the Log2 fold-changes. Among these genes were *Igfbp2*, *Sox9*, *Slc6a11*, and *Cd44*, which are known to contribute to astrocyte biology and astrocyte-neuron interactions[57–60] and are astrocyte-enriched relative to other cells in the striatum[19,20]. We also performed UpSet analysis of the unique overlap of the DEGs reported in those past studies and our data in

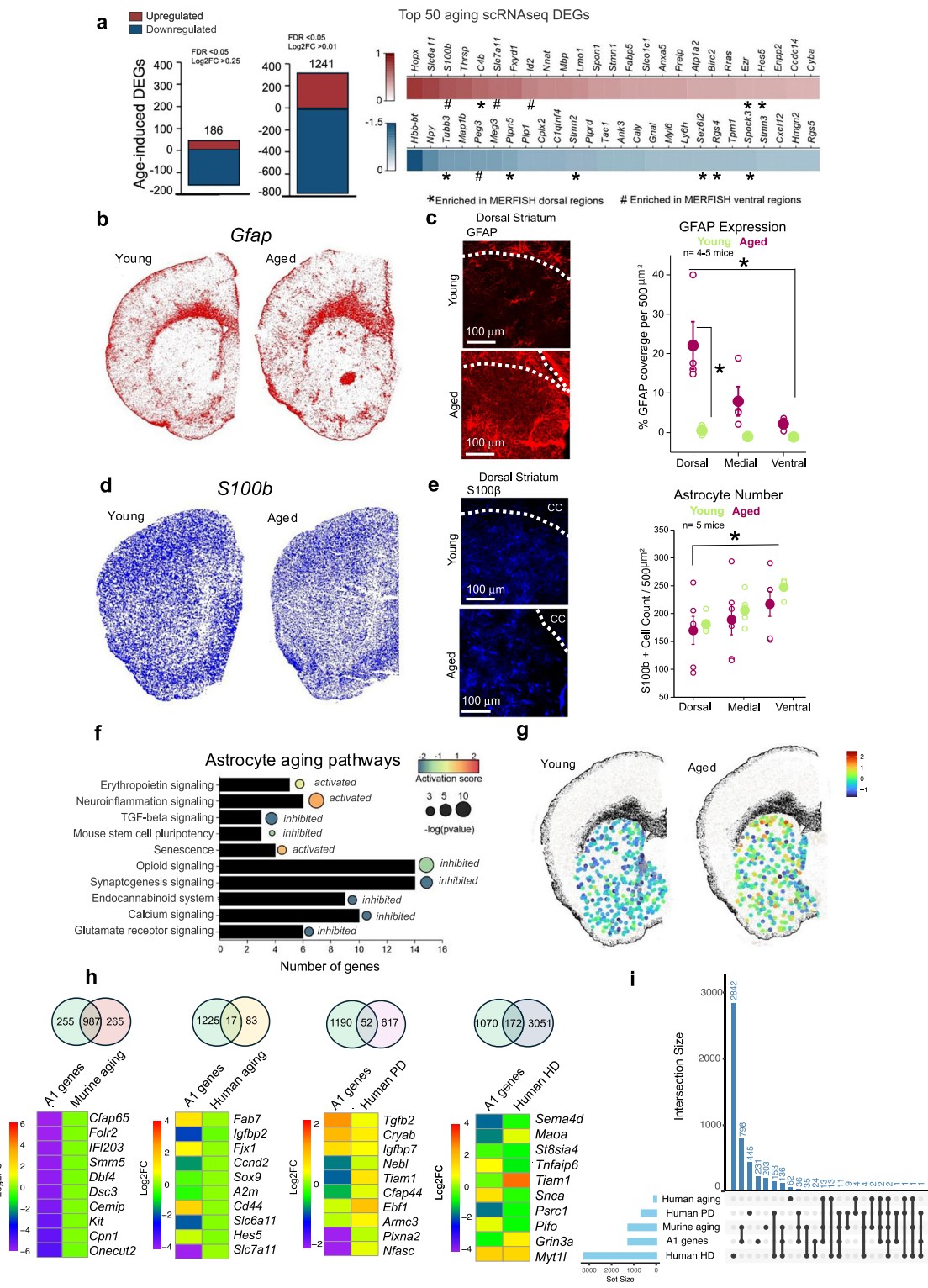

different combinations (Fig. 4i). This UpSet plot visualizes genes that exclusively overlap between different comparisons of the four datasets. In contrast to the overlap with pairwise comparisons, the number of uniquely shared genes across mouse striatal aging, human aging, human Huntington's disease, and human Parkinson's disease was much more modest (Fig. 4i). The number of unique genes shared across all datasets was only two (*CD44*, *EGFR*; Fig. 4i) and the degree of overlap between different combinations of conditions was variable and context specific (Fig. 4h, i; Supplementary Data 5). We

recognize that we have performed a conservative comparison, and further detailed evaluations of striatal astrocytes in human diseases are needed for more comprehensive evaluations to occur. However, the modest number of shared DEGs across conditions is consistent with recent insights that astrocytes change in diverse context-specific ways during disease[26,61], which implies disease and age-specific alterations that our data support (Fig. 4h, i). The richness and diversity of ways in which astrocytes change in disease has also been usefully reviewed[55].

**Fig. 4 | Molecular features of striatal astrocytes from aged mice. a** Top aging astrocyte markers were assessed using MAST differential expression, and the top 25 up and 25 downregulated transcripts are shown in the heatmap (see also Supplementary Data 5). Dorsal enriched genes are marked with an asterisk (*) and ventral enriched genes are marked with a hashtag (#). **b** Representative image of MERFISH RNA counts shows age increases in *Gfap* in the dorsal striatum. **c** Representative images of the dorsal striatum are shown for young and aged mice. GFAP coverage per 500 μm² in the dorsal striatum; across young and aged mice ($n$ = 4–5 mice; 2-way RM ANOVA with Bonferroni post hoc (*$p$ = 0.00032 (aged dorsal compared ventral) and *$p$ = 0.00029 (aged dorsal compared to young dorsal), data are presented as mean values ± SEM). **d** Representative image of MERFISH RNA counts shows *S100b* expression in the dorsal striatum. **e** Quantification of S100β + cells per 500 μm², in the dorsal, medial, and ventral striatum; across young and aged mice ($n$ = 4–5 mice; 2-way RM ANOVA with Bonferroni post hoc *$p$ = 0.036, data are presented as mean values ± SEM. **f** IPA analysis was performed on age-induced DEGs in striatal astrocytes, and each black bar indicates the number of genes per pathway, circle size indicates the −log($p$ value) (right-tailed Fisher's Exact Test), and circle color indicates the activation score number. **g** Aging gene score is calculated by the average expression levels of the top 20 differentially expressed genes in age on a single-cell level, subtracted by the aggregated expression of control feature sets. Each dot represents an astrocyte, and the color of the dot is the relative change in the aging score. **h** Heatmaps of the top 10 shared genes between our aging and A1 astrocyte subtype genes (Log2FC > 0.01) with mouse aging (Log2FC > 0.01), human striatal astrocyte aging, human striatal astrocyte Huntington's Disease, and human Parkinson's Disease using the DEG (see data availability and Supplementary Data 5). Venn diagrams visualize the total overlap across each gene set. **i** UpSet plot of the overlap of murine aging striatal data and the past studies.

## Astrocyte-neuron interactions change with age

To determine if astrocytes change their communication with neurons, which could potentially impact behavior or cognition, we employed CellChat[62], which infers communications by identifying over-expressed ligands or receptors in one cell group compared to other cell groups (Fig. 5). First, we evaluated the total number of interactions from major neuronal cell populations and either young or aged astrocytes (Fig. 5a). Our data suggest that aged astrocytes both send and receive less communication than young astrocytes. We also looked at an age and dorsal dominant astrocyte subtype (A1), compared to a young and ventral dominant astrocyte subtype (A4), and found A1 astrocytes send and receive less communication compared to A4 (Fig. 5a). We next investigated the specific ligand–receptor pairs from A1 or A4 to major neuronal populations (Fig. 5b). Of note in A1 astrocyte subtypes, *Nrxn-2* signaling from A1 astrocytes to neurons can no longer be detected in aged, especially in dopamine receptor expressing MSNs. From A4 astrocyte subtypes to D1 neurons, there is an increase in *Efna2*, *Flrt1*, and *Lama5* interactions with age (Fig. 5b). We next performed joint manifold learning and classification of the inferred communication networks based on their similarity across age. CellChat can identify the signaling networks with the most age-induced changes based on their Euclidean distance in the shared two-dimensional space. Larger distance implies larger difference of the communication networks between young cells (astrocyte + neurons) and old cells (astrocytes + neurons) (Fig. 5c). Many of the top communication networks are implicated in dysregulated immune function (*Pros1, Vista*) or endothelial function (*Jam, Pdgf, Gap*) (Fig. 5c). We took the top 10 interactomes by pathway distance and looked at their interaction across major neuronal cell populations and young/aged astrocytes or A1/A4 astrocytes (Fig. 5d). These top pathways converged on interaction between D1 MSNs, GABAergic interneurons (GIN) and astrocytes. In addition, A4 astrocytes receive more of these top interactome pathways than A1. Together, this suggests that astrocytes alone decline asymmetrically, and this has implications for their connections to some neurons more than others. Raw data for CellChat analysis of astrocytes, microglia, oligodendrocytes, and neurons are provided in Supplementary Data 4.

## Discussion

Aging represents a general degradation of functionality, with many systems and pathways declining simultaneously. The consensus in the field[57] is that aging is defined by twelve hallmarks: (1) genomic instability, (2) telomere attrition, (3) epigenetic alterations, (4) loss of proteostasis, (5) disabled macroautophagy, (6) deregulated nutrient-sensing, (7) mitochondrial dysfunction, (8) cellular senescence, (9) stem cell exhaustion, (10) altered intercellular communication, (11) chronic inflammation, and (12) dysbiosis. Our study focused on striatal astrocytes, which is germane to several but not all of the hallmarks of aging (e.g., neuroinflammation, autophagy, mitochondrial dysfunction, intercellular communication and senescence) and is relevant because multiple studies in mice and humans demonstrate that aging causes substantial transcriptomic changes to astrocytes and other glia —more so than in neurons[12–15]. Astrocytes are also diverse between brain regions and respond differently to age[32,33]. Based on these studies, we explored striatal astrocyte heterogeneity and how this was altered at two time points: 2 months of age, representing young adulthood, and 18 months of age, representing aging in mice[58]. We integrated several approaches to provide a view of striatal astrocytes and the impact of aging. We documented striatal astrocyte heterogeneity, their responses to aging, and mapped astrocytes within the structure of the striatum.

Increasing evidence demonstrates that astrocytes are heterogeneous throughout the brain, with circuit-specific roles and functions[19,20,59]. We studied astrocyte heterogeneity within the striatum in young and aged mice. Our IHC studies demonstrate that astrocytes exhibit different densities in the striatum, and that dorsal astrocytes have significantly higher levels of *Gfap*. We report seven astrocyte subtypes with robust, stable cluster-defining transcripts that were identified consistently across scRNAseq, MERFISH, and in the integrated datasets. The cross-platform validation and consistent emergence of these astrocyte subtypes through unsupervised stringent clustering bolsters this finding, but this does not obviate the need for additional studies of greater numbers of cells and with increased sequencing depth. MERFISH mapped the coordinates of striatal astrocytes and demonstrated that five of the seven astrocyte subtypes exhibited regional preference. Surprisingly, endothelial cells displayed similar levels of heterogeneity, but did not exhibit any clear spatial preferences. Although astrocytes and endothelial cells have intimate physical interactions at end feet, their spatial maps did not correlate. This may indicate that astrocyte spatial diversity is derived autonomously or from neuronal cues since neurons display striatal compartmentalization[60]. The realization that distinct astrocyte subtypes are anatomically allocated within the striatum recalls advances on astrocyte diversity between and within brain structures, including the striatum[16,19,20,23,61]. In future studies, it will be important to develop astrocyte subtype-selective mouse lines (expressing Cre recombinase or reporters) and viruses to study the distinct populations physiologically.

The impact of age on striatal astrocytes was not uniform, and astrocyte responses differed along the dorsal-ventral axis. Aging increased astrocyte GFAP coverage by 10-fold, but this change was largely confined to the dorsal striatum. Spatial transcriptomics demonstrated aging gene scores were higher in dorsal astrocytes than ventral further indicating dorsal striatal astrocytes are more impacted in the aged brain than ventral astrocytes. The higher density of ventral astrocytes may help buffer age-induced changes, or the different transcriptomes of these cells may contribute. Of note, astrocytes that have a higher density in the ventral striatum (A4) maintain predicted molecular interactomes with neurons that are lost in dorsal astrocytes (A1).

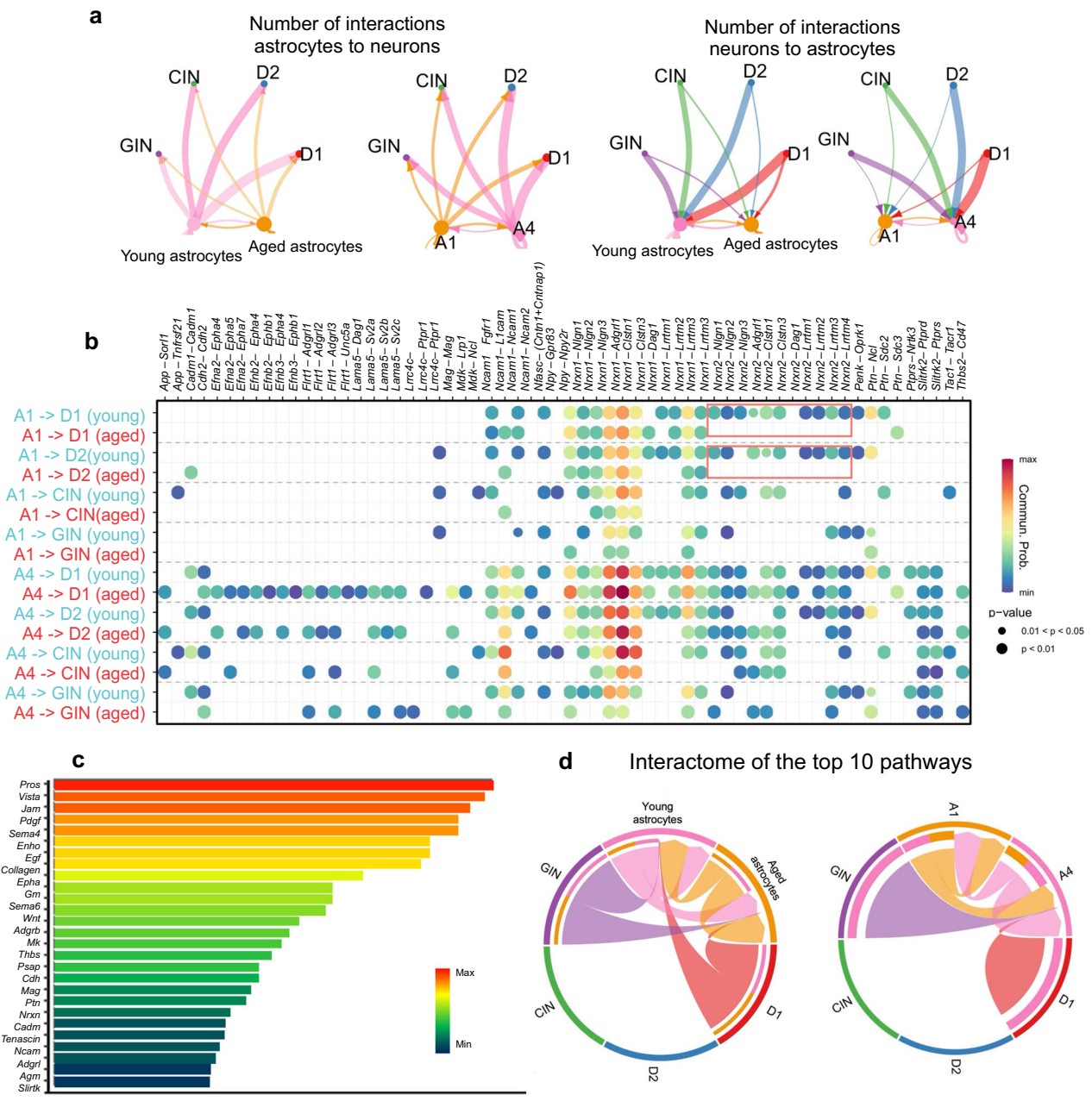

**Fig. 5 | Interactions between astrocytes and neurons in young and aged mice.** CellChat was used to investigate interactions in young and aged astrocytes across D1 expressing medium spiny neurons (D1), D2 expressing medium spiny neurons (D2), cholinergic interneurons (CIN), and GABAergic interneurons (GIN). **a** The number of interactions when astrocytes are the source/sender for young astrocytes and aged astrocytes highlights more interactions from young astrocytes to neurons. The number of interactions when astrocytes are the source/sender from astrocyte subtype A1 (aged and dorsal dominant subtype) and A4 (young and ventral dominant subtype) demonstrates that A4 astrocyte subtypes send more interactions to neurons than the A1 subtype. When neurons are the source/sender, both young astrocytes and A4 astrocyte subtypes receive more interactions than aged or A1 astrocytes, respectively. **b** Communication probability of significant pairs when A1 or A4 astrocyte subtypes are the source/sender to major neuronal

populations, split by astrocytes from young or aged mice. In A1 astrocyte subtypes, NRXN2 communication pairs are not detected in aged mice, while they are in young A1 astrocytes. *P* values are computed from a one-sided permutation test. **c** CellChat can identify the signaling networks with a larger (or smaller) difference based on their Euclidean distance in the shared two-dimensional space. Larger distance implies a larger difference in the communication networks between our two datasets (aged astrocytes and neurons vs young astrocytes and neurons) in terms of functional similarity. Here, we listed the top 25 pathways with the farthest (most significant) Euclidean distance. **d** We took the top 10 interactions (by Euclidean distance) and plotted their interactome across different cell types. This highlighted that interactions that change the most with age are interactions between GABAergic interneurons (GIN), D1 medium spiny neurons (D1), astrocytes (young and aged), and the A4 astrocyte subtype.

Our data show that in response to age, striatal astrocytes change to a more transcriptionally repressed state—with approximately 80 percent of DEGs decreasing in expression with age. This is separable compared to other cell types in the striatum but echoes striatal neurodegenerative diseases such as Huntington's disease[63]. This decline in

transcriptional complexity may also explain the decline in astrocyte heterogeneity observed, as well as the reduction in imputed ligand–receptor communication between astrocytes and neurons in aged mice. Of the transcripts that did increase with age in astrocytes, many are predicted to contribute to the neuroinflammation signaling

pathway. Together, this suggests astrocytes lose homeostatic functions, while switching to a reactive and neuroinflammatory state in the aging striatum[32,33]. Our findings recall recent work[58] and a review on the topic[64].

The MERFISH datasets revealed specific regional locations of striatal astrocyte subtypes, astrocyte-enriched genes, and aging DEGs. Astrocytes in the dorsal striatum had more robust age-induced transcriptional changes. This could be due to proximity to white matter tracts, as a recent study showed cells within and proximal to white matter tracts have more age-induced changes[58,65].

Our data build on past work demonstrating that aging differentially impacts cell classes. We show that even within one cell class–astrocytes–there are subtypes and regional differences in aging responses, which may contribute to specific behavioral tasks declining faster than others. Whether the changes in astrocyte molecular profiles contribute to changes in behavior is not demonstrated by our work, and this was not the goal of this study. The availability of our data will allow this topic to be tackled for striatum-dependent behaviors. However, we caution that aging overall is a general loss of functionality involving many major processes[57] and organ systems, all of which are not expected to be impacted directly by either the striatum or its astrocytes. Therefore, it seems unlikely that any one striatal astrocytic mechanism will have a profound direct effect on aging-related behaviors overall, but such mechanisms could affect striatal-dependent functions.

Exploring cellular diversity may lead to a deeper biological understanding of why select neural circuits are more susceptible to damage or decline. By demonstrating striatal astrocytes have diversity that contributes to their asymmetric aging profiles, our data provide evidence that cellular diversity and its alterations may have functional age-related consequences. However, we would be remiss if we did not explicitly state that additional studies with larger numbers of cells are needed across multiple ages and species, including humans. This is particularly the case for cells with low representation in our data. In addition, improved computational pipelines could be used to assess the data, as and when such newer bioinformatic methods continue to emerge. Thought leaders in the field have recently provided wise guidance on the interpretation of scRNAseq data, and such guidance should be incorporated in future studies before specific hypotheses are tested[66]. Whether astrocytic alterations, such as loss of essential normal homeostatic support, directly contribute to the decline of striatal-dependent functions will also need to be explored in future studies: our studies do not demonstrate that this is the case. In these regards, the single-cell and spatially resolved striatal astrocyte data reported here, along with past studies of pooled astrocytes from the striatum[33] and other brain areas[32] (see Introduction), provide the rationale for further exploratory and hypothesis-driven experiments focused on astrocytes, while carefully considering the strengths and limitations of the various multiomic methods[64].

Identifying causal and consequential roles for astrocytes in neural circuits and for behavior has taken considerable effort over many years[67]. Exploring how, or even if, astrocytes contribute to aging and age-related dysfunction is thus an important ongoing goal for the field, which will no doubt be advanced by improved methods and tools and the availability of additional, larger high-dimensional datasets, including from genetic, physiological, transcriptomic, and proteomic studies.

## Methods

All animal experiments were conducted in accordance with the National Institute of Health Guide for the Care and Use of Laboratory Animals and were approved by the Chancellor's Animal Research Committee at the University of California, Los Angeles (ARC-2017-090, ARC-2009-043). As per policy, mice were housed in vivarium holding rooms managed by the Division of Laboratory Animal Medicine at the University of California, Los Angeles (UCLA) with a 12 h light/dark cycle and *ad libitum* access to food and water. All animals were healthy with no obvious abnormal behavioral phenotypes, were not involved in previous studies, and were sacrificed during the light cycle. Wild-type C57BL/6NTac mice were generated from in-house breeding colonies or purchased from Taconic Biosciences.

### Striatal single-cell RNAseq (scRNAseq)

Single-cell RNAseq (scRNAseq) was performed identically on whole striata of adult and aged mice. All mice were euthanized with isoflurane and sacrificed at 9 am before brain collection, and prior to this there were no differences between mice in terms of housing, diet, handling, light-dark cycles, cage types, cage floor coverings, cage cleaning or anything else we could conjecture: as far as we can ascertain the mice for both ages were identically handled by us and the animal facility staff. Furthermore, the methods used to assess the cells were identical and performed within around 6 months, although for practical reasons (related to age) all the mice were not processed in parallel. Thus, KEL performed the molecular lab work for 2/4 young mice and 4/4 aged mice for scRNAseq, and XY performed the molecular lab work for 2/4 young mice and trained KEL. Male mice at 7–8 weeks old or 18 months old were anesthetized and decapitated. The brain was immediately dissected out and was sectioned on a vibratome (Microslicer DTK-Zero 1; Ted Pella, Inc.) into 400 μm slices in ice-cold artificial cerebrospinal fluid + trehalose (ACSF-T) (124 mM NaCl, 2.5 mM KCl, 1.2 mM NaH$_2$PO$_4$, 24 mM NaHCO$_3$, 5 mM HEPES, 13 mM glucose, 2 mM MgSO$_4$, 2 mM CaCl$_2$, and 14.6 mM trehalose with pH adjusted to 7.3–7.4) oxygenated with 95% O$_2$/5% CO$_2$ (carbogen). The slices containing the striatum were immediately transferred to an oxygenated recovery solution (93 mM N-methyl-D-glucamine, 2.5 mM KCl, 1.2 mM NaH$_2$PO$_4$, 30 mM NaHCO$_3$, 20 mM HEPES, 25 mM glucose, 10 mM MgSO$_4$, 0.5 mM CaCl$_2$, 5 mM sodium ascorbate, 2 mM thiourea, and 3 mM sodium pyruvate, 45 μM actinomycin D at a pH of 7.3–7.4) for 15 min on ice. The striatum was dissected out under a dissecting microscope in ice-cold ACSF-T and cut into small pieces (<1 mm in all dimensions). Tissue was then transferred to a Petri dish for digestion with ACSF-T containing 1 mg/ml pronase (Sigma-Aldrich, P6911) and 45 μM actinomycin D (Sigma-Aldrich, A1410) and incubated at 34 °C for 20 min while aerating with carbogen. The digested tissue was transferred to ice-cold oxygenated ACSF-T containing 1% fetal bovine serum and 3 μM actinomycin D. The tissue was dissociated sequentially by gentle trituration through glass Pasteur pipettes with polished tip openings of 500 μm, 300 μm, and 150 μm diameter. Actinomycin D was added to the recovery solution at 45 μM, the pronase solution at 45 μM, and the trituration solution at 3 μM to prevent stress-induced transcriptional alterations. To increase the yield of glial cells, filters with various pore sizes (70 μm, 40 μm, and 20 μm) were tested; 20 μm filters gave the highest yield and therefore were used. The dissociated cells were filtered through a 20 μm filter and washed with ice-cold ACSF-T. To remove myelin, the cell pellet was resuspended in phosphate-buffered saline (PBS) and processed with a debris removal kit (Miltenyi Biotec, 130-109-398). Cell density was counted, and isolated cells were diluted to 1000 cells/μl and processed with the 10X Genomics platform within 10 min. Single-cell libraries were generated and sequenced on the Illumina NextSeq500 sequencer. These methods have been described by us before[19,23,26], and the 2-month-old dataset was reanalyzed from our recent study[23].

### scRNAseq analysis

**Initial processing and quality control.** All scRNAseq data analysis in this study was performed by KEL. Sequence reads were initially processed and aligned to the mouse genome (mm10) using Cell-Ranger 3.0. For the subsequent analysis in R, we included striatal cells with at least 300 genes and retained genes expressed in more than 3 cells. Cells with mitochondrial percentages above 20% were

removed from the datasets. The initial dataset consisted of 64,836 cells with expression data for 21,019 genes. Young ($n = 4$) and Aged ($n = 4$) were integrated with Seurat's canonical correlation analysis in the following steps: after filtering (described above), each object was individually normalized, where transcript expression was normalized for each cell by the total expression, and multiplied by a scale factor of 10,000, and log-transformed, and then each object was scaled. Ambient RNA was removed from each dataset using DecontX. We report quality control metrics before and after filtering in Supplementary Fig. 13.

**Integration.** A list of the 8 objects was created, and the Seurat SelectIntegrationFeatures function was used with nfeatures = 3000. Anchors for the objects were built with FindIntegrationAnchors, using the features output from SelectIntegrationFeatures. Data were integrated using IntegrateData with the anchors from the output of FindIntegrationAnchors. The whole object was scaled, and the FindVariableFeatures function was applied. Next, PCA was carried out, and the top 30 principal components (PCs) were stored. Clusters were identified with the FindClusters() function by use of the shared nearest neighbor modularity optimization with a Louvian clustering resolution set to 0.08. Cell class clusters were then annotated based on the expression of cell lineage marker genes[68].

**Cell type assignment and subclustering.** Major cell classes of interest (astrocytes, endothelial cells, microglia, and oligodendrocytes) were further analyzed. Each clustered cell class was individually subset, CCA integrated as described above, scaled, normalized, PCA analysis performed, and shared nearest neighbor modularity optimization and Louvain clustering with resolution of 0.10–0.12 was performed for each glial cell class. This two-tiered analysis of glial cell clusters enabled us to investigate specifically the differences within each major glial cell class. We used SCPower to determine if our scRNAseq dataset had enough cells to identify the 7 astrocyte subclusters. (Supplementary Fig. 2). SCPower generates power curves by repeatedly subsampling the dataset to simulate different total cell numbers and measuring how often a given subcluster is detected. These power curves demonstrated we had enough power to detect the top 7 astrocyte subclusters, but not the smallest cluster, astrocyte subcluster 8 (Supplementary Fig. 2).

**Aging gene score calculation.** The aging gene score was calculated for astrocytes across the striatum using the module score function in Seurat. The top 20 DEGs across age (irrespective of astrocyte subtype) were selected for the module score. Seurat selects a set of control genes to serve as a baseline - these are selected based on the distribution of their expression level to match the distribution of the dataset. The number of control genes was set to 100, to ensure statistical robustness, and were randomly selected in bins based on their average expression level. To calculate the module score, Seurat subtracts the average module expression from the control expression, per individual cell.

**Comparison with past mouse astrocyte RNA sequencing datasets.** We examined astrocyte cluster-defining genes' expression across two mouse astrocyte RNA sequencing datasets. The first is from Chai et al., which has RiboTag RNA sequencing data from striatal astrocytes[20]. We found that 90.8% of cluster-defining genes are present in the RNAseq dataset from Chai et al.[20] We highlight 10 cluster-defining genes in Fig. 1f, and we visualized their FPKM expression from Chai et al.[20] in a grayscale heatmap in Fig. 1g. The second dataset to which we compared our astrocyte cluster-defining genes is from a single-cell RNA sequencing dataset[41] by Jin et al.[41] We found that 100% of cluster-defining genes are present in the astrocytes from the single-cell RNA-seq dataset from Jin et al.[41] We represent the raw count expression of

the highlighted astrocyte cluster-defining genes from the Jin et al.[41] dataset in the orange heatmap in Fig. 1g.

**Comparison with human datasets.** We compared our dataset with three available human datasets (aging (GSE46706 and GSE36192), PD (GSE157783), and HD (GSE242198)). We pulled age-induced DEGs across all astrocytes irrespective of subtype (murine aging), and from A1 astrocytes (A1 genes). For these comparisons, we used the cutoff 0.01 for the Log2FC to try and pull more significant DEGs for comparison. We used a Venn diagram through the website: https://bioinformatics.psb.ugent.be/webtools/Venn/. This allowed us to evaluate common genes across the four datasets. Given the differences in the methods and platforms used, these datasets were not statistically compared or normalized to one another, meaning that we compared the DEGs from the reported gene lists to our data.

## Immunohistochemistry (IHC)

For transcardiac perfusion, mice were euthanized with isoflurane and perfused with 10% buffered formalin (Fisher #SF100-20). Once all reflexes subsided, the abdominal cavity was opened, and heparin (50 units) was injected into the heart to prevent blood clotting. The animal was perfused with 30 mL ice-cold 0.1 M PBS followed by 30 mL 10% buffered formalin. After gentle removal from the skull, the brain was postfixed in 10% buffered formalin overnight at 4 °C. The tissue was cryoprotected in 30% sucrose PBS solution for at least 48 h at 4 °C until use. Forty micrometers coronal sections were prepared using a cryostat microtome (Leica) at 20 °C and processed for IHC. Sections were washed three times in 0.1 M PBS for 10 min each, and then incubated in a blocking solution containing 5% NGS in 0.1 M PBS with 0.2% Triton X-100 for 1 h at room temperature with agitation. Sections were then incubated with agitation in primary antibodies diluted in solution containing 5% NGS in 0.1 M PBS with 0.2% Triton X-100 overnight at 4 °C. The following primary antibodies were used: mouse anti-GFAP (1:1000; Millipore, MAB360), mouse anti-NeuN (1:500; Millipore MAB377), and rabbit anti-S100b (1:1000; Abcam ab41548). The next day, the sections were washed three times in 0.1 M PBS for 10 min each before incubation at room temperature for 2 h with secondary antibodies diluted in 5% NGS in 0.1 M PBS with 0.2% Triton X-100. Alexa conjugated (Molecular Probes) secondary antibodies were used at 1:500 dilution except streptavidin conjugated Alexa 647 at 1:250 dilution. The sections were rinsed three times in 0.1 M PBS for 10 min each before mounting on microscope slides in fluoromount-G. Fluorescent images were taken using UplanSApo 20× 0.85 NA, UplanFL 40× 1.30 NA oil immersion or PlanApo N 60× 1.45 NA oil immersion objective lens on a confocal laser-scanning microscope (FV10-ASW; Olympus). Laser settings were kept the same within each experiment. Images are shown as collapsed z-projections of sections. IHC images were processed with ImageJ (NIH) and IMARIS (Oxford Instruments). Cell counting was done on maximum intensity projections using the spots module in IMARIS.

## Dual in situ hybridization with RNAscope and IHC

For RNAscope and IHC, mice were euthanized with isoflurane, and brains were isolated. Fixed-frozen tissue was processed as described for IHC. Serial coronal sections (20 μm) containing striatum were prepared using a cryostat microtome (Leica) at −20 °C and mounted immediately onto glass slides. In situ hybridization was performed using Multiplex RNAscope (ACDBio #320851). Sections were washed at least for 15 min with 0.1 M PBS, and then incubated in 1X Target Retrieval Reagents (ACDBio #322000) for 5 min at 95–100 °C. After washing with ddH2O twice for 1 min each, they were dehydrated with 100% ethanol for 2 min and dried at RT. Then, the sections were incubated with Protease Pretreat solution (ACDBio #322340) for 30 min at 40 °C. The slides were washed with ddH2O twice for 1 min each and then incubated with probe(s) for 2 h at 40 °C. The following

probes were used: Mm-*Crym*-C3 (ACDBio #466131-C3), Mm-*Atp1a2*-C3 (ACDBio #569621-C3), Mm-*Cplx2*-C3 (ACDBio #573211-C3), Mm-*Aqp4*-C3 (ACDBio #417161-C3), Mm-*Rgs4*-C3 (ACDBio #467461-C3), Mm-*Gfap*-C3 (ACDBio #313211-C3), Mm-*Pdgfa*-C3 (ACDBio #411361-C3). The sections were incubated in AMP 1-FL for 30 min, AMP2-FL for 15 min, AMP3-FL for 30 min, and AMP4-FL for 15 min at 40 °C with washing in 1X Wash Buffer (ACDBio #310091) twice for 2 min each prior to the first incubation and in between incubations. All the incubations at 40 °C were performed in the HybEZ Hybridization System (ACDBio #310010). Slices were washed in 0.1 M PBS three times for 10 min each, followed by IHC that was performed as described above except with antibody dilutions. The following primary antibodies were used: guinea pig anti-NeuN (1:500, Synaptic system, #266004) and rabbit anti-S100ß (1:500; Abcam #ab41548). Images were obtained in the same way as for IHC and processed with ImageJ (NIH v2.1). For each gene of interest, the number of puncta and the fluorescence intensity were assessed in the soma of astrocytes and/or neurons.

## MERFISH

MERFISH was performed as described previously[69] according to the manufacturer's instructions (Vizgen). Briefly, mice were euthanized, and their brains were immediately placed in cold OCT. Embedded brains were then frozen at −20 °C and transferred to −80 °C for storage. Ten-micrometer coronal slices of the striatum were sectioned and placed upon a functionalized coverslip covered with fluorescent beads. Once adhered to the coverslip, the tissue was fixed (4% PFA in 1x PBS, 15 min, room temperature) followed by three washes with 1x PBS. After aspiration, 70% ethanol was added to permeabilize the tissue for 24 h. After a wash with formamide wash buffer (30% formamide in 2x saline sodium citrate (SSC)), the sample was incubated with a custom MERFISH probe library (Vizgen) and left to hybridize for 48 h. The sample was then washed and incubated at 47 °C with formamide wash buffer twice, and then the tissue was embedded in a polyacrylamide gel followed by incubation with tissue clearing solution (2x SSC, 2% SDS, 0.5% v/v Triton X-100, and 1:100 proteinase K) overnight at 37 °C. Then, the tissue was washed and hybridized for 15 min with the first hybridization buffer, which contained the readout probes associated with the first round of imaging. After washing, the coverslip was assembled into the imaging chamber and placed into the microscope for imaging. MERFISH imaging was performed on an automated Vizgen Alpha Instrument using imaging buffers, hybridization buffers, and parameter files provided by Vizgen. Briefly, the sample was loaded into a flow chamber connected to the Vizgen Alpha Instrument. First, a low-resolution mosaic image was acquired (DAPI channel) with a low magnification objective (10×). Then the microscope was switched to a high magnification objective (60×) and seven 1.5 μm z-stack images of each field of view position were generated in 750 nm, 650 nm, and 560 nm channels. A single image of the fiducial beads on the surface of the coverslip was acquired and used as a spatial reference. After each round of imaging, the readout probes were extinguished, and the sample was hybridized with another set of readout probes. This process was repeated until combinatorial FISH was completed. Raw data were decoded using the MERLIN pipeline (v.0.1.6, provided by Vizgen) with the codebook for the library (see Supplementary Data 2 for the gene panel).

## MERFISH data analysis

MERFISH identified segmented cells were filtered for quality control before subsequent analysis. Cells with less than 10 genes and/or less than 50 cubic micron³ volume were removed. To account for global differences in mRNA counts between samples, we normalized data to the total transcripts/cell for each sample. Data from all samples ($n = 4$) were merged into a single Seurat object for clustering and cell type annotation. Data were normalized by dividing gene counts/volume for each cell by total count/volume for that cell, multiplied by 10,000, and

log-transformed. Data were then scaled, and PCs were calculated on all 390 measured genes. The transcript expression was normalized for each cell by the total expression, multiplied by a scale factor of 10,000, and log-transformed. Next, PCA was carried out, and the top 15 PCs were stored. FindClusters() function by use of the shared nearest neighbor modularity optimization with a Louvian clustering resolution set to 0.08. Cell class clusters were then annotated based on the expression of cell lineage marker genes[68]. Cells that expressed markers for multiple cell types were removed. Plots generated using Seurat, ggplot2, scCustomize, and corrplot.

## CellChat data analysis

Astrocytes and neurons were subset from the larger single-cell sequencing dataset for interaction analysis using CellChat. CellChat identifies over-expressed ligands or receptors in one cell group and then identifies over-expressed ligand–receptor interactions if either ligand or receptor is over-expressed. The number of interactions across groups was assessed first. CellChat infers the biologically significant cell-cell communication by assigning each interaction with a probability value and performing a permutation test. CellChat models the probability of cell-cell communication by integrating gene expression with prior known knowledge of the interactions between signaling ligands, receptors, and their cofactors using the law of mass action. The law of mass action was then used to determine the communication probability from two main astrocyte subtypes (A1, A4) to major neuronal populations. CellChat performs joint manifold learning and classification of the inferred communication networks based on their functional and topological similarity across different conditions. We then ranked the communication networks by Euclidean distance to identify the top networks. Finally, we mapped the top 10 networks across young astrocytes, aged astrocytes, and major neuron cell populations: plus A1 astrocyte subtypes, A4 astrocyte subtypes, and major neuronal cell populations in a chord diagram.

## Data analysis, statistics, and reproducibility

Statistical tests were run in OriginPro 2020 (OriginLab, v.9.7). Summary data are presented as mean ± s.e.m. Sample sizes were not determined in advance and were based on past studies that are cited in the relevant sections of the manuscript and methods. Statistical tests were chosen as described below. All replicates were biological, not technical. For immunohistochemistry, the analyzer (KEL) was blind; for all other analyses, blinding was not done. For each set of data to be compared, we used OriginPro to determine whether the data were normally distributed or not. If they were normally distributed, we used parametric tests. If the data were not normally distributed, we used non-parametric tests. Paired or unpaired Student's $t$-test, Wilcoxon signed-rank test, or Mann–Whitney tests were used for statistical analyses with two samples (as appropriate). One-way ANOVA, two-way ANOVA, or repeated two-way ANOVA tests followed by Bonferroni's post hoc test were used for statistical analyses with more than three samples. Significant differences were defined as $P < 0.05$ and are indicated as such throughout.

## Reporting summary

Further information on research design is available in the Nature Portfolio Reporting Summary linked to this article.

# Data availability

scRNAseq datasets are available under GEO accession numbers GSE198027, GSE225741, and GSE226138. MERFISH datasets are available under GEO accession number GSE262083. Human striatal astrocyte aging is available under accession numbers GSE46706 and GSE36192, human striatal astrocyte Huntington's Disease is available under accession number GSE242198, and human Parkinson's Disease is available under accession number GSE157783. Source data for analyses

of scRNAseq, MERFISH, immunohistochemistry, and RNAscope are provided with this manuscript. Source data are provided with this paper.

## Code availability

Code is available at https://github.com/kaylinker/AgingAstrocytes.

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

## Acknowledgements

K.E.L. was supported by F32MH125598. This work was also supported by the NIH (DA047444, NS111583), the Allen Distinguished Investigator Award, a Paul G. Allen Frontiers Group advised grant of the Paul G. Allen Family Foundation, and the Ressler Family Foundation. D.P.S. was supported by NIMH/NIH (R01MH113743), NINDS/NIH (R01NS117533), NIA/NIH (RF1AG068281), the Massachusetts Life Sciences Center, and the Miriam and Sheldon G. Adelson Medical Research Foundation. V.D.L. was supported by BrightFocus Foundation (A2022006F) and Alzheimer's Association AARF-22-923219. We thank F. Endo and R. Kawaguchi for discussions on single-cell RNA sequencing analysis, and T. Faust for guidance in MERFISH data analysis. Thanks to Cai McCann (Broad Institute) for guidance on MERFISH.

## Author contributions

X.Y. performed single-cell RNA sequencing experiments for 2/4 young mice and trained K.E.L. K.E.L. performed the remaining scRNAseq and MERFISH experiments and analyzed all the data. V.D. aided K.E.L. with MERFISH experiments under the direction of D.P.S. M.O. performed and analyzed RNAscope experiments. B.S.K. and K.E.L. conceived and directed the experiments and planned the figures. B.S.K. and K.E.L. wrote the paper with help from the other coauthors. K.E.L. performed revisions to figures with guidance from B.S.K. The final text was revised by B.S.K., X.Y., and K.E.L. All authors commented.

## Competing interests

The authors declare no competing interests.
