## [Transparent Peer Review file · Nature Communications]

Aging alters regionally enriched striatal astrocytes

Corresponding Author: Professor Baljit Khakh

Version 0:

Reviewer comments:

Reviewer #1

(Remarks to the Author)

This revised manuscript retains all of the important aspects of this study, but now contextualizes is more appropriately in the literature. The authors are to be commended for their continued provision of clarity for the reviewers. This is a nice paper with data that is going to be prove useful for the field.

Specific comments:

1. the additional in situ for cluster-specific genes is much appreciated, and does a great deal to further validate these sequencing data. In addition the statement from the authors clarifying that “cluster-defining” genes are not synonymous with unique “marker” genes in the classical sense is pertinent, and understood by this reviewer. I look forward to a review on this misinterpretation in the literature from the authors in the future. It is an important point, and one that is often overlooked.

2. the addition of accession numbers requested by Reviewer 2 (missed by this reviewer) is much appreciated.

I have no further comments that would improve the manuscript further. I applaud the authors, and look forward to future work from the lab.

Comments to Reviewers

There were no additional comments from the Reviewers for us to address.